# pH-Responsive Lipid Nanoparticles Achieve Efficient mRNA Transfection in Brain Capillary Endothelial Cells

**DOI:** 10.3390/pharmaceutics14081560

**Published:** 2022-07-27

**Authors:** Yu Sakurai, Himeka Watanabe, Kazuma Nishio, Kohei Hashimoto, Atsuki Harada, Masaki Gomi, Masayoshi Suzuki, Ryotaro Oyama, Takumi Handa, Risa Sato, Hina Takeuchi, Ryoga Taira, Kenta Tezuka, Kota Tange, Yuta Nakai, Hidetaka Akita, Yasuo Uchida

**Affiliations:** 1Laboratory of DDS Design and Drug Disposition, Graduate School of Pharmaceutical Sciences, Tohoku University, 6-3 Aoba, Aramaki, Aoba-ku, Sendai 980-8578, Japan; himeka.watanabe.q7@dc.tohoku.ac.jp (H.W.); kazuma.nishio.t7@dc.tohoku.ac.jp (K.N.); kohei.hashimoto.t7@dc.tohoku.ac.jp (K.H.); atsuki.harada.s3@dc.tohoku.ac.jp (A.H.); masayoshi.suzuki.s8@dc.tohoku.ac.jp (M.S.); takumi.handa.q2@dc.tohoku.ac.jp (T.H.); risa.sato.t8@dc.tohoku.ac.jp (R.S.); hina.takeuchi.t5@dc.tohoku.ac.jp (H.T.); ryoga.taira.q2@dc.tohoku.ac.jp (R.T.); kenta.tezuka.r5@dc.tohoku.ac.jp (K.T.); 2Laboratory of DDS Design and Drug Disposition, Graduate School of Pharmaceutical Sciences, Chiba University, 1-8-1 Inohana, Chuo-ku, Chiba 260-0856, Japan; gomi.masaki.t6@dc.tohoku.ac.jp (M.G.); oyama.ryotaro.p3@dc.tohoku.ac.jp (R.O.); 3DDS Research Laboratory, NOF CORPORATION, 3-3 Chidori-cho, Kawasaki-ku, Kawasaki 210-0865, Japan; kota_tange@nof.co.jp (K.T.); yuta_nakai@nof.co.jp (Y.N.)

**Keywords:** lipid nanoparticle, ssPalm, mRNA transfection, blood–brain barrier, hCMEC/D3 cells, cell toxicity, SWATH-MS, translation, chaperonin-containing TCP-1

## Abstract

The blood–brain barrier (BBB), which is comprised of brain capillary endothelial cells, plays a pivotal role in the transport of drugs from the blood to the brain. Therefore, an analysis of proteins in the endothelial cells, such as transporters and tight junction proteins, which contribute to BBB function, is important for the development of therapeutics for the treatment of brain diseases. However, gene transfection into the vascular endothelial cells of the BBB is fraught with difficulties, even in vitro. We report herein on the development of lipid nanoparticles (LNPs), in which mRNA is encapsulated in a nano-sized capsule composed of a pH-activated and reductive environment-responsive lipid-like material (ssPalm). We evaluated the efficiency of mRNA delivery into non-polarized human brain capillary endothelial cells, hCMEC/D3 cells. The ssPalm LNPs permitted marker genes (GFP) to be transferred into nearly 100% of the cells, with low toxicity in higher concentration. A proteomic analysis indicated that the ssPalm-LNP had less effect on global cell signaling pathways than a Lipofectamine MessengerMAX/GFP-encoding mRNA complex (LFN), a commercially available transfection reagent, even at higher mRNA concentrations.

## 1. Introduction

Various cells are currently available as in vitro models for the BBB model. However, none of them completely reflect in vivo brain capillary endothelial cells. One example, a human brain capillary endothelial cell line (hCMEC/D3), has a smaller claudin-5 expression than isolated brain capillaries and therefore results in the formulation of weaker tight junctions [1]. Another example, human-induced pluripotent stem cell-derived brain microvascular endothelial cells (hiPS-BMECs), show significantly lower expression levels of the multidrug resistance protein (MDR1/P-gp), and therefore, they lack drug efflux activity [2]. Therefore, the transfection of genes that can complement the protein to the level of that in the in vitro model cells would be a powerful tool for establishing a more valid BBB model that can mimic in vivo. However, the transfection of brain capillary endothelial cells has been reported to be difficult in some studies. It has been reported that, in primary cerebral vascular endothelial cells, less than 5% of the cells are transfected with plasmid DNA when a commercially available transfection reagent, LyoVec is used [3]. Such a low transfection efficiency has also been reported for hCMEC/D3 cells. For example, when hCMEC/D3 cells were transduced with a P-gp fusion gene and the green fluorescent protein (GFP) using the Lentivirus vector, the transfection efficiency was only approximately 10% [4]. One of the reasons for the low transfection efficiency is attributed to poor cellular uptake. In fact, it has been reported that the percentage of cells that took up double-stranded DNA as an agonist of the toll-like receptor was less than half that of hepatocytes and macrophages using a cationic polymer [5]. While there is no doubt that hCMEC/D3 cells are potentially useful as an in vitro BBB model, a reliable method for introducing specific genes remains to be developed. Lipofectamine is widely used, but a major drawback to its use is that the cationic liposomes contained in it produce unintended cytotoxicity. In human endothelium-derived cells (HUVECs), lipofectamine 2000 has been reported to inhibit cell proliferation, reduce the expression of various proteins, and cause an unfolded protein response (UPR) [6]. We here hypothesize that the use of an uncharged neutral nanoparticle would be more feasible for use in conjunction with endothelial cells.

To achieve a more efficient gene delivery into human brain capillary endothelial cells, hCMEC/D3, we developed lipid nanoparticles (LNPs) that contain encapsulated mRNA [7]. This LNP was designed to satisfy two fundamental properties, namely, high biocompatibility and high intracellular auto-degradability. To achieve this, we developed an ss-cleavable pH-activated lipid-like material (ssPalm), which was equipped with tertiary amines that develops a positive charge in response to the acidic pH and a disulfide-bonded unit that undergoes molecular disintegration in the intracellular reductive environment (Appendix A). When the ssPalm molecules are reconstituted into the LNPs (ssPalm-LNPs), the entire LNP had an apparent acid dissociation constant (p*Ka*) of around 6.3. Therefore, under physiological conditions, the ssPalm-LNP would behave as a neutral nanoparticle. However, in the acidic environment of endosomes, they would become positively charged and then fuse with the negatively charged cell membrane. The fusion between LNP and cell membrane in acidic endosome led to membrane disruption and subsequent endosomal escape of LNP to cytosol. The ssPalm also reacts with reducing agents such as the glutathione present in the cell and disintegrates, thus releasing the loaded nucleic acid into the cytoplasm [7]. Controlling the intracellular dynamics of encapsulated nucleic acids through such multi-step actions of these functional units enabled mRNA to be efficiently introduced into cells that are generally considered to be relatively resistant to transfection [8]. In this study, we attempted to apply the ssPalm-LNP for the transfection of mRNA into hCMEC/D3 cells, for which introducing genes has been difficult.

Regarding the evaluation of adverse effects after the transfection, a comprehensive analysis of the protein expression level would be useful. The SWATH-MS (sequential window acquisition of all theoretical fragment ion spectra mass spectrometry) method is one of the recently developed quantitative proteomics methods, whose quantitative accuracy is higher than previous comprehensive proteomics technology [9]. Using this methodology, we compared the variation in the level of protein expression after the transfection with the ssPalm-LNP.

Based on these analyses, we report herein on the advantage of using the ssPalm-LNP in terms of achieving a high mRNA transfection to the non-polarized human brain capillary cells, hCMEC/D3, with low adverse effects.

## 2. Materials and Methods

### 2.1. Cell Culture

The detail in the hCMEC/D3 culture procedure was almost the same as previously reported but slightly modified [1,10,11]. The hCMEC/D3 cells were cultured in Endo-GRO complete Media Kit (Merck-Millipore, Burlington, MA, USA) on the plate coated with Cultrex Rat type-I collagen (R&D Systems, Minneapolis, MN, USA) at 37 °C under a 5% CO_2_ atmosphere for 3–4 days, and maintained until used for mRNA transfection.

### 2.2. Preparation of ssPalm-LNP Encapsulating GFP-Encoding mRNA

The GFP-encoding mRNA modified with 5-methoxyuridine was obtained from TriLink Biotech (San Diego, CA, USA). The ssPalm-LNP using SS-OP (provided by NOF CORPORATION), 1,2-dioleoyol-*sn*-glycerophosphocholine (DOPC, NOF CORPORATION), cholesterol (Chol, Sigma-Aldrich, Burlington, MA, USA) and poly(ethylene) glycol (average molecular weight 2000)-1,2-dimyrisotyl-*sn*-gycerol (PEG-DMG, NOF CORPORATION) was prepared as previously [12,13]. The structures of these lipids are depicted in Appendix A. We formulated mRNA into LNP using ethanol dilution methods, in which lipid molecules/mRNA was spontaneously assembled by gradual decrease in ethanol concentration, based on previously published methodology [14]. Briefly, 3 μg of GFP-encoding mRNA, dissolved in 45 μL of 25 mM malic buffer (30 mM NaCl, pH 3.0), was gradually added to the lipid mixture (total 131.5 nmol (SS-OP/DOPC/Chol 52.5/7.5/35.0) with 3.95 nmol PEG-DMG) in 30.26 μL of ethanol. The solution was then further diluted with 1 mL of 20 mM 2-(N-morpholino)ethanesulfonic acid) (MES) buffer (pH 5.5, 30 mM NaCl) under vigorous mixing with a vortex mixer. The resulting mixture was diluted with 3 mL of MES buffer and then ultrafiltered with an Amicon Ultra-4 (Merck-Millipore, molecular weight cut off: 100 kDa). The concentrate was again ultrafiltered after the dilution with phosphate-buffered saline without Mg^2+^ and Ca^2+^ (PBS). The obtained LNP was characterized using a ZetaSizer Pro (Malvern Panalytical, Malvern, UK). The recovery rate and the encapsulation efficiency of the mRNA were determined by a RiboGreen assay, as previously reported [12,13]. To label the LNP with near-infrared fluorescence, 1,1′-dioctadecyl-3,3,3′,3′-tetramethylindodicarbocyanine (DiD) was added to the lipid mixture at 0.5 mol% to total lipid moles before mixing it with the mRNA solution.

### 2.3. Preparation of Lipofectamine/GFP-Encoding mRNA Complex (LFN)

Lipofectamine MessengerMAX (ThermoFisher Scientific, Waltham, CA, USA) was used as a control. According to the manufacturers’ protocol, 1.5 μL of Lipofectamine Messenger MAX was incubated with 1000 ng of GFP-encoding mRNA for 10 min.

### 2.4. Flow Cytometry Analysis of Cellular Uptake and Gene Expression

hCMEC/D3 cells were plated on a 12-well plate at a density of 7.5 × 10^4^ cells/well 24 h before the addition of the ssPalm LNP and LFN. The cells were incubated with the DiD-labeled ssPalm-LNP at a dose of 300 ng mRNA in 750 μL of culture medium with 5% fetal bovine serum (0.4 μg/mL) for 2 h (cellular uptake) at 37 °C. To prepare the DiD-labeled LFN, DiD was first mixed with the Lipofectamine MessengerMAX solution so that the final fluorescence intensity in the LFN solution was the same as that for the ssPalm-LNP solution.

To measure GFP expression, the cells were exposed to the ssPalm-LNP solution that had been diluted with a culture medium at an mRNA concentration of 300 ng in 750 μL (0.4 μg/mL) for 16 h. The dilution volume of the ssPalm-LNP was adjusted based on the mRNA recovery ratio measured by the RiboGreen assay result in each experiment. As a control, the LFN was then added to cells cultured in a 12-well plate and incubated for 16 h. The incubated cells were then washed twice with 1.0 mL of PBS and then trypsinized. The obtained cells were suspended in PBS with 0.5% bovine serum albumin and 0.1% sodium azide and analyzed using a NovoCyte (Agilent Technology, Santa Clara, CA, USA).

### 2.5. Observation of Cellular Uptake and Gene Expression with Con-Focal Laser Scanning Microscopy

hCMEC/D3 cells were plated onto a glass-based 8-well chamber plate, which had been pre-coated with type-I collagen, at a density of 500 cells/well 48 h before the addition of the ssPalm-LNP or LFN. The cells were incubated with the ssPalm-LNP at a concentration of 300 ng/200 μL (0.4 μg/mL) for 48 h at 37 °C in the presence of 5% fetal bovine serum. As a control treatment, cells were exposed to LFN at the same dose as the ssPalm LNP. Both the ssPalm-LNP and the LFN complex were first diluted in 200 μL of culture medium. Nuclei were stained by a 10-min incubation in 1.0 μg/mL Hoechst33342. After washing with 200 μL of PBS, cells were observed by a Nikon C1 confocal laser scanning microscope system (Nikon, Tokyo, Japan).

### 2.6. Evaluation of Cytotoxicity in mRNA Transfection Using ssPalm-LNP and LFN in hCMEC/D3 Cells

The hCMEC/D3 cells were treated with the ssPalm-LNP containing GFP-encoding mRNA, and the LFN with 1250, 2500, and 6250 ng of GFP-mRNA in 1 mL (1.25, 2.5, and 6.25 μg/mL) for 48 h at 37 °C in 5% CO_2_ in the presence of 5% fetal bovine serum, and observed under a microscope.

### 2.7. SWATH-MS Analysis in hCMEC/D3 Cells Treated with ssPalm LNP and LFN

The ssPalm-LNP and the LFN were treated with hCMEC/D3 cells at a concentration of 2.5 μg/mL for 48 h at 37 °C in 5% CO_2_. The SWATH-MS analysis was then performed for the whole cell lysate of hCMEC/D3 cells, as previously described [15,16]. Briefly, after a 48-h treatment, 6 well plates of cells were placed on ice, and the cell surface was immediately washed with ice-cold PBS. A denaturing buffer (7 M guanidium hydrochloride, 0.5 M Tris–HCl (pH 8.5), 10 mM EDTA) was directly added to the cell surface in order to prepare a whole cell lysate. The dissolved cells were subjected to up-and-down strokes in a 27G × 1/2 syringe (Terumo, Tokyo, Japan) to completely lyse the cells. The protein concentration of the whole cell lysate was determined by a BCA assay (Thermo Fisher Scientific Inc., Waltham, MA, USA). A 50 µg sample of protein in the whole cell lysate was reduced, S-carboxymethylated, and purified by methanol-chloroform precipitation. The precipitate was solubilized in a urea buffer containing 0.05% ProteaseMax surfactant (Promega, Madison, WI, USA), and the proteins were digested with lysyl endopeptidase (Lys-C, Wako Pure Chemical Industries, Osaka, Japan) at an enzyme/substrate ratio of 1:100 for 3 h at 30 °C. The resulting Lys-C digested proteins were then digested with TPCK-treated trypsin (Promega, Madison, WI, USA) at an enzyme/substrate ratio of 1:100 for 16 h at 37 °C. After a C18 clean-up, the digested protein sample was injected into a NanoLC Ultra system (Eksigent Technologies, Dublin, CA, USA) coupled with an electrospray-ionization Triple TOF 5600 mass spectrometer (SCIEX, Framingham, MA, USA), which was set up for single direct injection, and analyzed by SWATH-MS acquisition. The details of measurement and subsequent data analysis have been described previously [17,18]. Finally, the relative expression levels among the control, ssPalm-LNP-treated, and LFN-treated groups were determined for all the quantified proteins. Benjamini–Hochberg adjusted *p* values were calculated, and based on a cutoff of 0.05, the issue of whether there is a significant difference in protein expression level between the groups was determined.

## 3. Results

### 3.1. The mRNA Introduced by the ssPalm-LNP Is Homogeneously Translated in hCMEC/D3 Cells

The ssPalm-LNPs that were manually prepared by the vortex method had an average particle size of 106.0 ± 15.0 nm, and polydispersity index was 0.09 ± 0.02 in PBS. The ζ-potential of −1.5 ± 1.0 mV in 10 mM HEPES buffer (pH 7.4). The distribution in particle size is shown in Appendix A and indicates that particles with homogenous size distribution were successfully prepared. The mRNA recovery rate was 91.7 ± 12.8%, and the encapsulation efficiency was 86.5 ± 6.1%. This was comparable to a previous report using in vitro a transcribed mRNA encoding luciferase [12]. Considering LFN, size distribution is also homogenous (polydispersity index 0.17 ± 0.03) but slightly large (diameter 340.5 ± 2.5 nm) in PBS (Appendix A), indicating colloidal formulation of LFN/mRNA complex is also stable even in an isotonic condition.

GFP expression was initially observed by microscopy (Figure 1). Most of the cells treated with the ssPalm-LNPs expressed GFP (Figure 1 left), while only a small fraction of cells was positive in the LFN-treated group (Figure 1 right). To quantitatively evaluate the heterogeneity in gene expression, GFP fluorescence was analyzed by flow cytometry (Figure 2). The mean for the GFP fluorescence intensity in the ssPalm-LNP-treated cells was significantly higher than that in the LFN-treated cells. Of note, the ssPalm-LNP induced GFP expression in a majority (95.0 ± 2.1%) of the cells (Figure 2C), consistent with the microscopic observations. On the other hand, the LFN treatment resulted in the GFP expression in a small fraction of cells (0.43 ± 0.27%). In conclusion, the ssPalm-LNPs showed a drastically higher transfection in hCMEC/D3 cells compared to LFN.

### 3.2. Higher Amount of the ssPalm-LNP Internalizes hCMEC/D3 Cells Than LFN

Cellular uptake was also evaluated to determine the reasons for the ssPalm-LNPs being highly expressed in hCMEC/D3 cells. We found that the more homogenous the particles (Figure 3A), the higher the uptake of the ssPalm-LNPs (Figure 3B) in comparison with LFN (Figure 3B). Additionally, the % of DiD-positive cells were also higher in ssPalm-LNP treatment group (Figure 3C).

### 3.3. mRNA Transfection into hCMEC/D3 Cells by ssPalm-LNP Is Significantly Less Cytotoxic Than LFN

To observe the cytotoxicity of the ssPalm-LNP and LFN, hCMEC/D3 cells were observed after treatment at a higher concentration than the above concentrations that were used in these gene expression experiments. In the LFN-treated group, some cells were detached at higher concentrations (Figure 4). The morphology of the cells also differed from that of non-treated cells. At an mRNA concentration of 6.25 μg/mL, these morphological changes were more prominent. In contrast, in the ssPalm-LNP treatment, no change in cell morphology and no detachment of cells were observed, even at the highest concentration (Figure 4).

### 3.4. ssPalm LNP Has Significantly Fewer Adverse Effects than LFN

To evaluate possible effects on off-target proteins, hCMEC/D3 cells were analyzed after being exposed to a higher concentration of the ssPalm-LNP and LFN than used in the gene transfection experiment. A total of 1899 proteins were quantified by SWATH-MS analysis in the control, ssPalm-LNP-, and LFN-treated groups (Appendix A). The levels of expression of four typical proteins in cerebral vascular endothelial cells (PECAM1, GLUT1, MDR1, and ZO-1) were unchanged among these three groups (Appendix A). Out of 1899 proteins, the significantly up- and down-regulated proteins in the ssPalm-LNP-treated group against the control group were 14 and 26 molecules, respectively (Figure 5). In contrast, the proteins that were significantly up- and down-regulated in the LFN-treated group compared to the control group were 7 and 77 molecules, respectively (Figure 5). This suggests that the LFN treatment results in a more prominent perturbation (mainly decreased) in protein expression, whereas this was minimal in the case of the ssPalm-LNP treatment.

In the above experiments, unlike LFN, the ssPalm-LNPs did not cause cytotoxicity. To understand the molecular mechanisms that are involved in this difference, proteins with significant differences in expression between the LFN and control groups, but not between the ssPalm and control groups, were extracted. As a result, 67 proteins were extracted (Appendix A). These were analyzed by a “String functional protein association network” analysis, and as the top two clusters, translation-related proteins and chaperonin-containing TCP-1 (CCT) (Figure 6) were then identified. As translation-related proteins, the levels of expression of RPS3, RPL11, RPL27, EIF2S3, EIF3F, EFTUD2, and HSPD1 proteins were significantly decreased by the LFN treatment by 1.17-, 1.17-, 1.28-, 1.20-, 1.27-, 1.10-, and 1.12-fold, respectively, whereas no significant reduction was observed in ssPalm-LNP-treated group (Figure 6). As CCTs, the levels of expression of the TCP1, CCT2, CCT3, and CCT4 proteins were significantly reduced by the LFN treatment by 1.15-, 1.13-, 1.10-, and 1.16-fold, respectively, but not significantly in the ssPalm-LNP-treated group (Figure 6).

## 4. Discussion

In the present study, we reported that mRNA-loaded ssPalm-LNPs can be used to efficiently transfect hCMEC/D3 cells with a marker gene GFP without any obvious toxicity. Although previous reports have not provided a clear reason for the low transfection efficiency in BBB-derived endothelial cells, the present results suggest that one major factor is the low uptake of cationic substrates (Figure 3). Commonly used cancer cells and mouse embryo fibroblasts exhibited a much higher uptake of cationic nanoparticles compared to neutral ones [7,20,21]. This high uptake of cationic substances is thought to be, at least in part, due to interactions with negatively charged proteoglycans on the outer surface of the cell membrane [22]. Although the exact mechanism is unclear, it is possible that this uptake pathway might not be available in hCMEC/D3 cells. On the other hand, the uptake of neutral LNPs, including ssPalm-LNPs, is attributed to the formation of complexes of LNPs with Apoproteins (Apos) in the culture medium or in the biological fluid, and apolipoprotein receptors such as low-density lipoprotein receptors (LDLR) subsequently recognize these complexes [23,24,25]; it has been reported that LDLR and low-density lipoprotein receptor-related protein (LRP), a receptor for Apos, are expressed in BMEC [26,27]. The current SWATH-MS results showed the expression of LRP2, LRP8, and various apoproteins such as apoproteins A-I, B, E, and L3 (Appendix A), although the amounts are not currently known. Taken together, the uptake of ssPalm-LNP may have been much higher than that of LFN due to the interaction of these receptors in hCMEC/D3 with medium-derived Apos that are adsorbed on the LNPs. In addition, it should be noted that the mRNA localization and whole structure of these nanoparticles were completely different: mRNA would be encapsulated into lipid molecules (LNP), or mRNA would be attached to the cationic surface of liposomes (LFN). Actually, the size of the ssPalm LNP was smaller than LFN (106.0 nm vs. 340.5 nm). The effect of the difference in the size and structure on the gene transfection efficacy should be taken into account.

The ssPalm-LNPs did not alter the morphology of the hCMEC/D3 cells (Figure 4), even though cells were treated with the ssPalm-LNP at 15.6-fold higher concentrations than that for the controls where >95% of the cells were transfected (Figure 1 and Figure 2). Furthermore, compared to the LFN, the ssPalm-LNPs had less adverse effects on cellular protein expression (Figure 5). These data suggest that the ssPalm-LNPs developed in this study are superior for mRNA delivery to brain capillary endothelial cells in terms of cytotoxicity and off-target effects. It should also be noted that the levels of expression of many proteins were significantly down-regulated as the result of the LFN treatment (Figure 5). This is consistent with a report showing that the expression of various proteins was reduced in Lipofectamine 2000-treated HUVEC cells [6]. This may be attributed to the significant reduction in protein translation, such as ribosomal proteins and chaperonin (HSPD1/HSP60 and CCT/TRiC) in the LFN-treatment group (Figure 6).

When misfolded proteins accumulate in the cell, the unfolded protein response (UPR) is activated at the endoplasmic reticulum (ER) to reduce the accumulation of misfolded proteins. If the UPR fails to restore the ER to normality, ER stress can promote apoptosis [6]. In HUVEC cells, lipofectamine 2000 causes UPR and suppresses cell proliferation [6]. CCT/TRiC and HSPD1/HSP60 are two important chaperonins that interact with misfolded proteins to prevent misfolding and aggregation and facilitate correct folding. These levels of expression of these proteins were reduced in the case of the LFN treatment in hCMEC/D3 cells (Figure 6). This may have resulted in an over-accumulation of misfolded proteins in the cells, causing cell death. Since ssPalm-LNPs do not adversely affect the expression of molecules that are involved in protein translation and chaperonins (HSPD1/HSP60 and CCT/TRiC), they would be expected to be safe materials for future in vivo gene delivery to brain capillary endothelial cells.

The currently reported mRNA transfection technology could be applied to the expression of various genes. We demonstrated the successful encapsulation of mRNA molecules ranging from 850 bases to more than 4.5 kilobases regardless of the presence or absence of chemical modification (i.e., N1-Methylpseudouridine) [7]. Additionally, the introducible proteins are not limited to cytosolic proteins such as GFP: membrane proteins such as transporters and tight junction-related proteins can also be expressed. Further, although we demonstrate herein the efficient gene transfection with monocultured, non-polarized brain capillary endothelial cells, the mRNA transfection by the ssPalm-LNP is expected to be applied for multiplexed culture (endothelial cells, astrocytes, pericytes, and related cells), which are currently used as in vivo BBB model [28,29] since the LNP can transfect mRNA even in complete medium containing serum.

## 5. Conclusions

We report herein that the ssPalm-LNP is a promising carrier for efficiently transporting mRNA into the non-polarized human brain capillary endothelial cells, hCMEC/D3. The transfection efficiency of the ssPalm LNP was found to be much higher than that of a commercially available transfection reagent Lipofectamine MessengerMAX. Treatment with high concentrations of the ssPalm-LNP did not induce cytotoxicity in comparison with LFN. Further, a SWATH-MS analysis revealed that the exposure of hCMEC/D3 cells to the ssPalm-LNP had only minimal effects on cellular proteins. These results indicate that the ssPalm-LNP represents a potent tool for elucidating the functional, biological chrematistics of the brain endothelium by the transfection of a gene of interest.

## Figures and Tables

**Figure 1 pharmaceutics-14-01560-f001:**
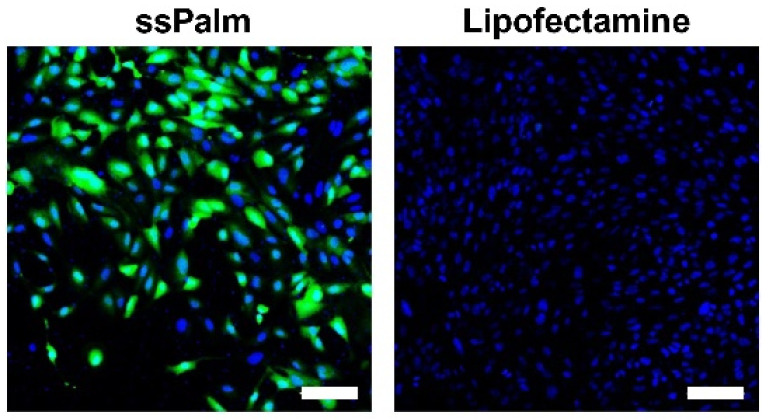
GFP expression by mRNA transfection using the ssPalm LNP or LFN. The GFP expression in hCMEC/D3 observed after mRNA transfection at a concentration of 0.4 μg/mL. Nuclei (Hoechst33342) and GFP were depicted in blue and green, respectively. Scale bars: 100 μm.

**Figure 2 pharmaceutics-14-01560-f002:**
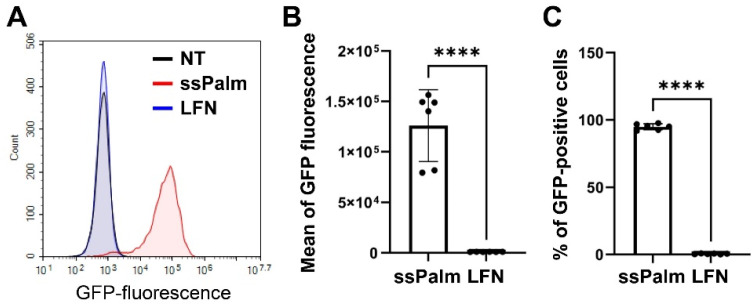
Flow cytometry analysis of GFP expression after mRNA transfection by the ssPalm-LNP or LFN. The GFP expression in hCMEC/D3 cells after mRNA transfection at a concentration of 0.4 μg/mL. (**A**) The representative histogram of the GFP expression in hCMEC/D3 cells. Black, blue, and red lines indicate non-treatment (NT), LFN treatment, and ssPalm-LNP treatment, respectively. (**B**,**C**) The mean GFP fluorescence intensity (**B**) and the percentage of GFP-positive hCMEC/D3 cells (**C**) were analyzed in several independent experiments. Data represent the mean ± standard deviation. Student’s t-test was performed between ssPalm and LFN. ****: *p* value < 0.001.

**Figure 3 pharmaceutics-14-01560-f003:**
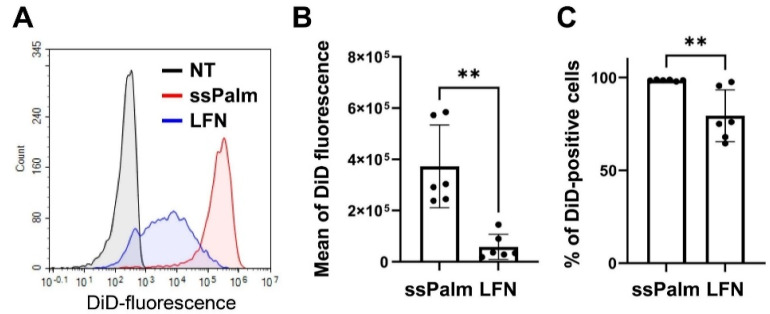
Cellular uptake of the ssPalm LNP or LFN. The cellular uptake of the ssPalm-LNP and LFN by hCMEC/D3 cells was measured by flow cytometry. (**A**) Representative histogram of the uptake of DiD-labeled ssPalm-LNP and LFN by hCMEC/D3 cells at a concentration of 0.4 μg/mL. Black, blue, and red lines indicate the non-treated (NT), the LFN-treated, and the ssPalm-LNP-treated group, respectively. (**B**,**C**) The mean of DiD fluorescence intensity (**B**) and the percentage of DiD-positive hCMEC/D3 cells (**C**) were calculated in several independent experiments. Data represent mean ± standard deviation. Student’s *t*-test was performed between ssPalm and LFN. **: *p* value < 0.01.

**Figure 4 pharmaceutics-14-01560-f004:**
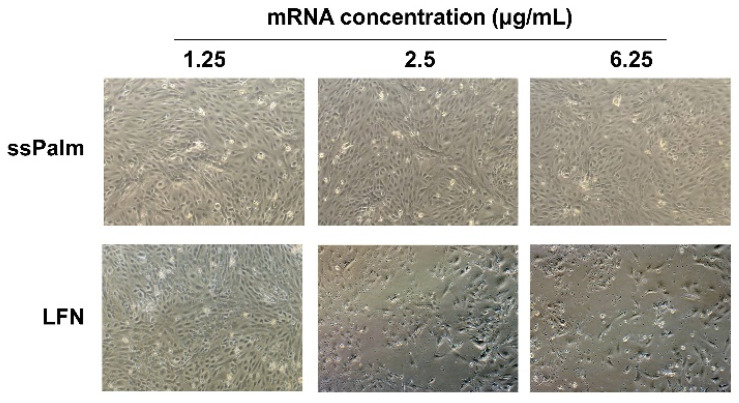
Microscopic observation of hCMEC/D3 cells after the treatment with ssPalm-LNP and LFN. The ssPalm-LNP and the LFN were treated with hCMEC/D3 cells for 48 h at the indicated concentrations.

**Figure 5 pharmaceutics-14-01560-f005:**
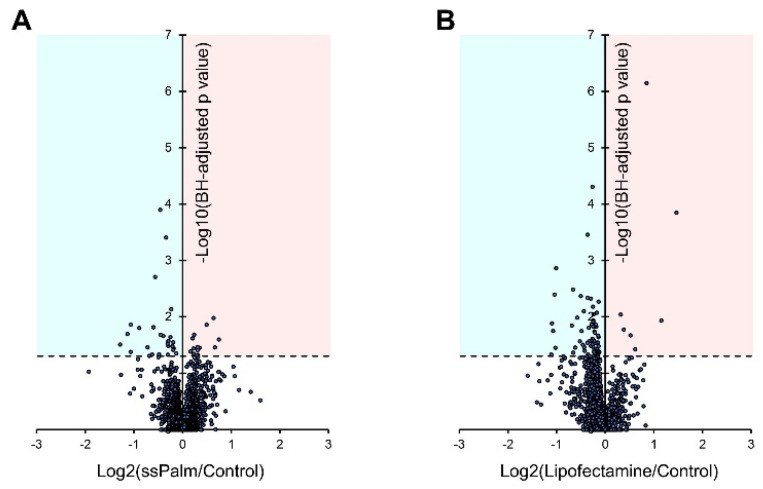
Volcano plot representation for the protein expression levels of all the quantified proteins by SWATH-MS analysis. The ssPalm-LNP (**A**) and LFN (**B**) were treated with hCMEC/D3 cells for 48 h at a concentration of 2.5 μg/mL. Then, the SWATH-MS analysis was performed for the whole cell lysate of hCMEC/D3 cells. The levels of expression of all the quantified proteins were compared with those in the control group (*n* = 5–9). X-axis represents the log2 values for the fold changes in protein expression levels compared with the control group. Y-axis represents the minus log10 values of the Benjamini–Hochberg (BH) adjusted *p* values for the differences between two groups. Blue and red areas represent the down- and up-regulated proteins with *p* values less than 0.05.

**Figure 6 pharmaceutics-14-01560-f006:**
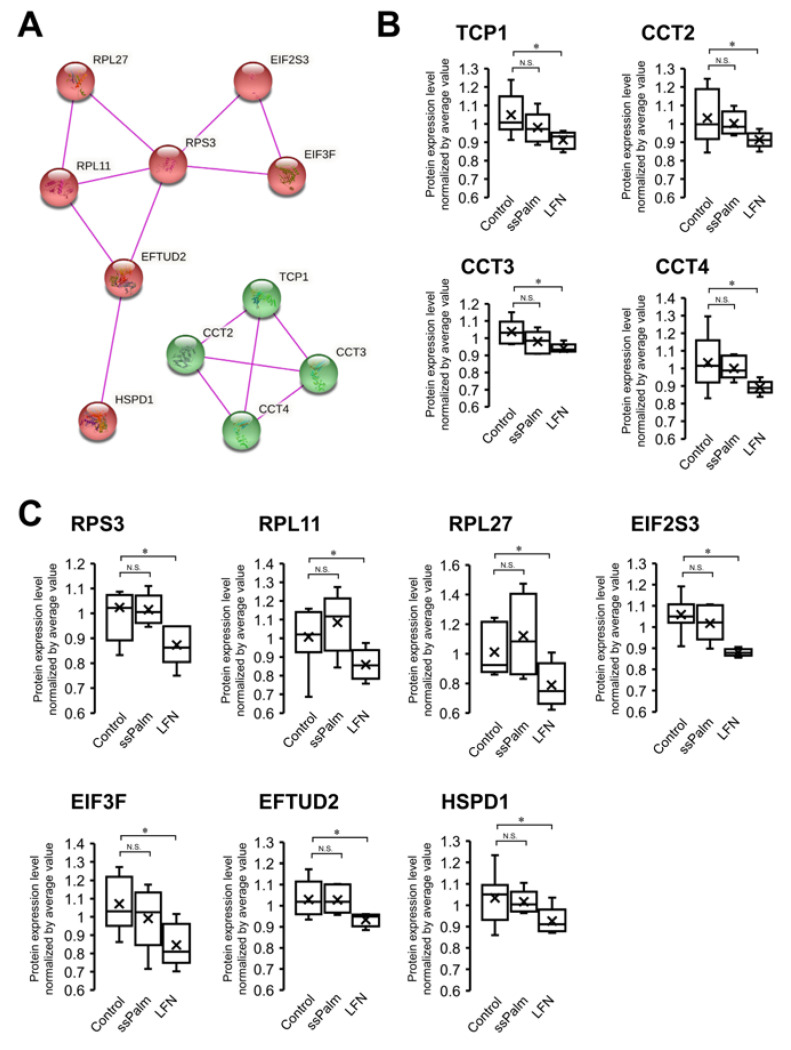
Top 2 clusters that are down-regulated in the LFN group but not in the ssPalm group. (**A**) 67 proteins whose expression levels were significantly changed in the LFN-treated group but not in the ssPalm-LNP-treated group (Appendix A analyzed using the STRING Database (https://string-db.org/ accessed on 2 May 2022) to visualize functional protein association networks [19] with an MCL clustering mode based exclusively on experimentally determined interactions. For the top 2 clusters, the interaction maps obtained on the String website are shown in this figure. The first cluster consists of translation-related proteins (red-colored nodes), and the second cluster consists of chaperonin-containing TCP-1 (CCT) proteins (green-colored nodes). (**B**,**C**) the levels of expression of CCT (**B**) and translation-related proteins (**C**) were compared among the control, ssPalm-LNP-treated group, and LFN-treated groups (*n* = 5–9). The band inside the box represents the median, and the bottom and top of the box indicate the first and third quartiles, respectively. Whiskers indicate the minimum and maximum values of the protein levels. X plots show the average in each group. * the Benjamini–Hochberg adjusted *p* value < 0.05 was significantly down-regulated compared to control group. N.S.—not significantly different (Benjamini–Hochberg adjusted *p* value > 0.05).

## Data Availability

The data presented in this study are available from the corresponding authors on reasonable request.

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
