# Peer review of "pH-Responsive Lipid Nanoparticles Achieve Efficient mRNA Transfection in Brain Capillary Endothelial Cells"

_pharmaceutics, 2022, doi:10.3390/pharmaceutics14081560_

Round 1

Reviewer 1 Report

The submitted manuscript "ssPalm-based lipid nanoparticles achieve efficient mRNA transfection in brain capillary endothelial cells" is a nice work, carried out carefully, and clearly illustrated and discussed. The proposed lipid nanoparticles represents a potent tool for elucidating the functional, biological chrematistics of the brain endothelium and the BBB by the transfection of a gene of interest.

Minor points:

1) The reviewer would like the authors not to use the abbreviation “ssPalm” in the title. In addition, even if the abbreviations of several terms are defined in the abstract, it could be nice to also do so in the first occurrence of the term in the main text.

2) Materials and methods, line101: could you please give more information about the celle line and the culture conditions or at least cite a publication on that ? Is it necessary to say that the cell line was kindly provided by Dr Pierre-Olivier Couraud, since you indicate it in the the Acknoledgements line 396 ?

3) Could you please explain why the p values were based on a cutoff of 0.05 ?

4) Legend to figure 6; it could be obvious to change the sentence Line 305: …were analyzed in a String functional protein association network analysis with an MCL clustering mode based on experimentally determined interactions…

By

…were analyzed using the STRING Database (https://string-db.org/) to visualize functional protein association networks [ref x] with an MCL clustering mode based on experimentally determined interactions only.

[ref x] Szklarczyk D*, Gable AL*, Nastou KC, Lyon D, Kirsch R, Pyysalo S, Doncheva NT, Legeay M, Fang T, Bork P, Jensen LJ, von Mering C. The STRING database in 2021: customizable protein–protein networks, and functional characterization of user-uploaded gene/measurement sets . Nucleic Acids Res. 2021 Jan 8;49(D1):D605-12.PubMed

Reviewer 2 Report

The authors state that using hCMEC/D3 cells as an in vitro BBB model is ineffective due to the leaky nature of these cells (due to a lack of claudin 5) and that using hiPS-BMECs is also problematic due to the lack of P-gp. They suggest that transfecting cells with genes that would complement protein levels in vitro would produce a BBB model that could mimic the in vivo state. They discuss the use of lipid nanoparticles for the delivery of model nucleic acids to hCMEC/D3 cells. It has been well established over recent years that monocultures as BBB models are ineffective, most groups have moved towards triple culture models incorporating both pericytes and astrocytes (and on occasion neurons). These models have demonstrated increased TEER values, ZO1 and claudin-5 expression and low paracellular transport. Therefore the continued use of a monoculture BBB model seems dated and of little utility.

Reviewer 3 Report

The authors present an interesting study examining a new and novel means of transfection of mRNA transcripts such as to alter the genomic content of a target cell population. Specifically, the authors highlight the apparent difficulty of transfecting cells of the blood-brain barrier with existing methods purported to increase likelihood of cell death in addition to poor overall transduction. The authors contrast a lipid-based nanocarrier they developed with lipofectamine reagent for the delivery of GFP to an immortalised cell line of the cerebral endothelium. In comparing the two methods, the authors highlight a near 100% transfection rate with the lipid-based nanocarrier in contrast to the much lower, significantly more toxic lipofectamine-based approach. Moreover, the authors highlight the lipid-based nanocarrier perturbed the proteomic profile of the cells to a far lesser degree than that of the lipofectamine-based counterpart, further underscoring the advantages of such a system over the currently and widely available systems.

In reviewing the manuscript however I had a number of concerns. The following should be addressed by the author in advance of any resubmission.

1.       The authors highlight the difficulty in transfecting cells of the blood brain barrier in the introduction however in this reviewers experience this is not as difficult as the authors are suggesting. Some examples of poor transfection are given, but to say that ‘generally’ it is difficult is inaccurate. The authors must revise this writing.

2.       Figure 2C indicates that almost 100% of cells expressed GFP when the mRNA was delivered by the lipid nanocarrier. In Figure 1, an image of the cells having been treated with the same experimental conditions is provided. The image and the flow cytometry data do not quite align. There are clearly a significant number of DAPI stained cell not expressing GFP in Figure 1, indicating that the measurement by flow cytometry is inaccurate. The authors must revise this data in advance of any resubmission.

3.       Similarly, in Figure 1 an image of cells treated with lipofectamine is given, with a comparable cell population present as compared to those treated with the lipid-based nanocarrier. However, in later data, the authors suggest there is a high cytotoxic effect when lipofectamine is used. Similar to the previous points, these data do not quite align with one another. The authors must address this in any resubmission.

4.       The authors highlight the reduced toxicity of the lipid-based nanocarrier in contrast to the lipofectamine-based alternatives, with brightfield microscopy images presented as evidence. Why did the authors not perform a quantitative means of assessing viability? The brightfield images, while additive, are a basic assessment of this index and this aspect of the study needs to be re-evaluated to support the claims being made. The authors must address this in any resubmission.  

5.       Can the authors clarify is the lipid-based nanocarrier treated cells were treated in serum-free conditions?

6.       Are there any limitation to the nanocarriers themselves? Is the size/charge distribution uniform/within certain limits? Do they uptake the same concentration of GFP each? Is there a limit on the length or nature of the mRNA that can be used in conjunction with the material? There is not much characterisation of the nanocarriers themselves, so it would be interesting to hear the authors investigations into such.

7.       Did the authors perform any long term studies to see how stable the transfection is? Did the cell eventually clear the GFP? Was it any quicker than the lipofectamine counterpart?

Round 2

Reviewer 2 Report

Unfortunately even after some minor changes to the manuscript and after reading the authors comments I do not think this article is appropriate for publication in Pharmaceutics. I feel it is of little utility or interest to researchers working in this area.

Reviewer 3 Report

The authors have clarified aspects of their studies in their rebuttal, and in some instances the issues raised were addressed. I do have concerns however over the study design, both in terms of the hypothesis and the granular detail included in this study. As mentioned previously, I do not believe transfection in brain endothelial cells is as big a issue as is implied in the context of the study, and in putting forward a method which aims to address this 'issue' there are aspects of this study which are incomplete and not comprehensive enough in their detail to truly convince this is a replacement for existing strategies.